# Dyeing Improvement and Stability of Antibacterial Properties in Chitosan-Modified Cotton and Polyamide 6,6 Fabrics

**DOI:** 10.3390/jfb14100524

**Published:** 2023-10-18

**Authors:** Marta Piccioni, Roberta Peila, Alessio Varesano, Claudia Vineis

**Affiliations:** CNR-STIIMA (National Research Council—Institute of Intelligent Industrial Technologies and Systems for Advanced Manufacturing), Corso Giuseppe Pella 16, 13900 Biella, Italy; marta.piccioni@stiima.cnr.it (M.P.); alessio.varesano@stiima.cnr.it (A.V.); claudia.vineis@stiima.cnr.it (C.V.)

**Keywords:** textiles, cotton, polyamide 6,6, chitosan, antibacterial, dyeing

## Abstract

Cotton and polyamide 6,6 fabrics coated with chitosan, a natural biopolymer, have been tested against two different bacteria strains: *Staphylococcus aureus* as Gram-positive bacterium and *Escherichia coli* as Gram-negative bacterium. Using the ASTM standard method (Standard Test Method for Determining the Antimicrobial Activity of Antimicrobial Agents Under Dynamic Contact Conditions) for antibacterial testing, the treated fabrics is contacted for 1 h with the bacterial inoculum, the present study aims to investigate the possibility to reach interesting results considering shorter contact times. Moreover, the antibacterial activity of chitosan-treated fibers dyed with a natural dye, *Carmine Red*, was evaluated since chitosan has an interesting property that favors the attachment of the dye to the fiber (cross-linking ability). Finally, fabric samples were tested after washing cycles to verify the resistance of the dye and if the antibacterial property was maintained.

## 1. Introduction

Fabrics and protective clothes used in schools, hotels, hospitals, nursing homes, and crowded public areas can benefit from antimicrobial finishing. Antimicrobial finishes are an important market need to prevent a reduction in mechanical strength and the formation of unpleasant odors in athletic wear, intimate apparel, underwear, socks, upholstery, and hospital linen [1].

Other effects due to the biodeterioration of fabrics are the formation of stains and discoloration, which is of interest for almost all types of fibers.

Natural fibers, like cotton, are generally more susceptible to biodeterioration than synthetic fibers because their porous hydrophilic structure retains water, oxygen, and nutrients, providing an ideal environment for microbial growth [2].

Generally, the antibacterial finishes should be non-toxic to the textile user, and allergy, cytotoxicity, irritation or sensitization must be avoided.

The treatment should maintain the textiles’ quality, handle or appearance, and it must have excellent fastness for use, mainly for repeated laundering. Also, the application method should be simple, easily implementable in the finishing process, and environmentally friendly [2].

Often, antimicrobial agents for textiles are synthetic, including metals (e.g., silver), metal oxides (e.g., zinc oxide) or salts, polymers (e.g., polypyrrole), and chemicals (e.g., quaternary ammonium compounds, synthetic azo dyes, triclosan). Most of these agents are toxic to humans and are not environmentally friendly.

Polypyrrole is an example of a synthetic antimicrobial agent that is safe for human skin with precaution for respiratory tissue [3], but the monomer (before synthesis) is toxic.

Quaternary ammonium compounds (QACs) are another example of antimicrobial agents of synthetic origin. These are widely used in industrial applications because they are effective against Gram-positive and Gram-negative bacteria, fungi, and certain viruses [4]. Despite the effectiveness of QACs, they present a disadvantage, namely, the poor fastness of the treatment due to the fast leaching from the textile for the lack of chemical or physical bonding [2].

Silver is an antimicrobial metal widely used in textiles. However, there are some concerns about its toxicity, particularly when used as nanoparticles [5,6].

The possible toxic effects of some of these agents on human beings are reported in Table 1 [7].

In view of these environmental and ecological concerns, there has been a focus on the research and development of new antimicrobial compounds of natural origin in the last decades.

Plants have gained interest as a source of natural antimicrobials. In 2016, Katewaraphorn et al. [14] investigated the antibacterial activity of cotton fabrics treated with a leaf extract of *Psidium Guajava* containing phenolic compounds.

Strong antimicrobial properties have been found in flavonoids, quinonoids, terpenoids, and tannins extracted from different parts of plants such as roots, bark, leaves, and flowers [15].

However, chitosan and its derivatives appear to be the most effective natural antimicrobial agent on the market [2].

In fact, the natural polysaccharides used for the functional finishing of textiles are abundantly available as waste products and are of an eco-friendly nature [7].

Chitosan, 2-amino-2-deoxy-(1→4)-d-glucopyranan, is undoubtedly one of the more promising multifunctional polymers among textile finishing agents.

It is chemically composed of glucosamine and N-acetylglucosamine units linked by 1–4 glucosidic bonds [16] and has unique properties, such as biodegradability, non-toxicity, and antimicrobial activity [17].

Chitosan is derived from the deacetylation process by chitin, which is the second most abundant biopolymer in the world after cellulose. Chitin is a component of the shells of crustaceans and constitutes the exoskeleton of insects as well as the wall of fungi [18,19].

Chitosan shows antibacterial activity against both Gram-positive and Gram-negative bacteria, thanks to its combined bactericidal and bacteriostatic action. Its property mainly includes four mechanisms [17]: it causes damage to microbial DNA, as a blocking agent of oxygen and nutrients in the bacteria cell, it can bind to cationic metals (calcium, magnesium) and nutrients essential for the microorganism, and it causes the loss of cytoplasmic intracellular components necessary for cell survival.

The first two mechanisms are associated with the chitosan antibacterial activity against Gram-negative bacteria, while the third and fourth dominate the antibacterial activity against Gram-positive bacteria.

All antibacterial mechanisms described are due to the fact that chitosan molecules carry positively charged amine groups and thus, in turn, have an electrostatic interaction with negatively charged cell membranes of microorganisms [20]. This electrostatic interaction leads to bacterial death.

Moreover, chitosan has the intrinsic property of acting as a cross linker, favoring the attachment of natural dyes to the fibers [21]. This is due to the chemical structure of chitin that is similar to cellulose, with a hydroxyl group on each monomer replaced with an acetylamine group [19,22]. The different structures are reported in Figure 1. It is an important property because most natural dyes generally lack substantivity for fibers, especially fibers such as cotton [23].

In a recent study [24], a double-layered chitosan coating was cured on cotton fabric to serve as a biomordant and form a protective layer on it. Through a second chitosan layer cured on the dyed fabric via the cross-linking method, the washing fastness of the cotton fabric dyed with sodium copper chlorophyllin can be improved from 3 to 5 (according to the standard method EN ISO 105-C06).

In 2021, Verma et al. [25] investigated the effect of biopolymer and dyeing treatment with natural dye on the functional properties (i.e., antibacterial and UV protection) of cotton fabric. It was found that the chitosan-treated onion skin-dyed cotton fabric showed 97.20% and 98.03% reduction in the growth of *Escherichia coli* and *Staphylococcus aureus* bacteria, respectively, and provided high UV protection.

In the present work, cotton and polyamide 6,6 fabrics were coated with a solution of chitosan to be tested as antibacterial fabrics against Gram-negative and Gram-positive bacteria.

The novelty of the present research work is related to the study of the contact time between the bacterial inoculum and the treated fabrics in order to evaluate how the antibacterial action of the chitosan-coated fibers changes, particularly after dyeing and washing.

The existing standard test methods [26] suggest using 1 h of contact time between the bacterial inoculum and antibacterial specimen. The aim of this work is to evaluate if shorter contact times can be more selective in evaluating the antibacterial performances of fabrics.

Chitosan has been selected because it quickly exerts a high antibacterial action in filtration [27], it is stable in water [22], and is widely proposed as a dyeing enhancer in textile finishing due to its cross-linking properties [23].

Since durability to repeated washing is the major challenge for the practical use of antimicrobial textiles [28], fabrics coated with chitosan have been subjected to several washing cycles to evaluate the resistance of the dye and if the antibacterial property is maintained.

## 2. Materials and Methods

### 2.1. Bacteria Strains and Fabrics

In this study, the antibacterial activity of fabrics coated with a chitosan solution was evaluated using model strains obtained from the American Type Culture Collection (ATCC): Gram-positive bacteria, *Staphylococcus aureus* ATCC 6538 and Gram-negative bacteria, *Escherichia coli* ATCC 11229 (supplied by Biogenetics Diagnostics Srl, Ponte San Nicolò, Italy). The media used in the ASTM E 2149-13 (Standard Test Method for Determining the Antimicrobial Activity of Antimicrobial Agents Under Dynamic Contact Conditions) was yeast extract agar (nutrient broth agar).

The adjacent cotton fabric (plain weave fabric suitable for ISO 105-F02, mass per unit area 110.75 g/m^2^ determined in accordance with ISO 3801, supplied by Testfabrics Inc., West Pittston, PA, USA) and adjacent polyamide 6,6 fabric (plain weave fabric suitable for ISO 105-F03, mass per unit area 130.0 g/m^2^ determined in accordance with ISO 3801, supplied by Testfabrics Inc., USA) were chosen to be used as textile substrates coated with chitosan for evaluating the antibacterial activity of the biopolymer. Fabrics were used as received from the supplier.

### 2.2. Preparation of the Chitosan Solution

A 2% *w*/*w* low molecular weight chitosan (50–190 kDa supplied by Sigma-Aldrich, Milan, Italy) in a 2% *w*/*v* glacial acetic acid solution was prepared. The solution was shaken for 7 h at room temperature.

Ten grams of cotton and polyamide 6,6 fabrics were dipped into the solution overnight to promote the adsorption of chitosan on the fabrics.

The impregnated fabrics were manually padded to reach a 90% wet pick up.

The fabrics were eventually placed in an oven at 95 °C for 3 min and at 150 °C for another 3 min to allow for the reaction between the chitosan and fabrics.

Finally, cotton and polyamide 6,6 materials were stored at 20 °C and 65% relative humidity at least 24 h before testing and further treatments.

### 2.3. Dyeing of Treated Fabrics

The chitosan-treated fabrics were dyed with the natural colorant *Carmine Red* (E120, C.I. Natural Red 4, C.I. 75470, supplied by Aromata Group srl, Bresso, Italy) which is largely soluble in water. Due to its high chemical and biological stability, it is used in textiles, pharmaceuticals, cosmetics, food, and beverages [29].

In the dyeing process, a 4% dye concentration was employed. The bath ratio was set at 1:20 (i.e., 10 g of fabric in 200 mL bath). For comparison, both the chitosan-treated and the untreated fabrics were dyed.

An Ahiba Nuance Top Speed II (Datacolor Italia srl, Giussano, Italy) dyeing machine was used. The working temperature was set at 100 °C with a 1 °C/min heating rate and left for 1 h. No additive was used during the dyeing.

At the end of the process, the chitosan-treated cotton fabric was homogeneously dyed, while the untreated cotton fabric maintained its natural white color (Figure 2).

The same experiment was realized on the synthetic fabric and in this case, as shown in Figure 3, only the chitosan-treated polyamide 6,6 fabric was dyed.

### 2.4. Washing Fastness

The washing fastness of the dyed samples was evaluated according to the standard method EN ISO 105-CO6 (2010) [30].

A water solution with a non-ionic detergent, Triton, was prepared at a concentration of 4 g/L. Dyed samples (10 cm by 5 cm) were cut and sewn with a sample of multifiber fabric (supplied by Testfabrics Inc., West Pittston, PA, USA) of the same size to verify if, during washing, the dye was released and absorbed by another type of fiber.

The samples were placed into a 150 mL bath together with ten stainless steel balls.

The temperature was raised to 40 °C and left for 30 min. The fabrics were then rinsed under tap water and dried in an oven at 37 °C.

It was observed that in both types of fibers, natural and synthetic, the color remained absorbed on the fabric even after washing. This was due to the property of chitosan which acts as a “bridge” between the dye and the fabric, ensuring washing fastness.

### 2.5. Characterizations

The natural and synthetic fabrics coated with dyed and washed chitosan were characterized by scanning electron microscopy (SEM), colorimetric analysis, water contact angle, and FT-IR analysis.

Characterizations were carried out on chitosan-treated fibers and dyed chitosan-treated fibers, and subjected to washing cycles.

The surface morphology of these fabrics was examined by SEM with a Zeiss EVO 10 (by Carl Zeiss AG, Oberkochen, Germany) scanning electron microscope at an acceleration voltage of 15 KV, a current probe of 400 pA, and a working distance of about 30 mm. The fabric samples were mounted on aluminum specimen stubs and sputter coated with a 20 nm thick gold layer in rarefied argon (20 Pa) using an Emitech K550 Sputter Coater with a current of 20 mA for 180 s before SEM observations.

Colorimetric analysis was performed with a Datacolor SF 600 X Spectralflash (Datacolor Italia srl, Italy) with CIE standard illuminant D65, 10°. Color coordinates and ∆*E* CIELab values were registered on the specimen of the dyed cotton and polyamide 6,6 fabrics with a size of about 5.0 × 5.0 cm. The reference samples for ∆*E* values were the untreated cotton and polyamide 6,6 fibers. The CIELab ∆*E* values were calculated according to Equation (1):(1)$$∆E=\sqrt{{\left(L-L_{0}\right)}^{2}+{\left(a-a_{0}\right)}^{2}+{\left(b-b_{0}\right)}^{2}}$$
where *L*_0_, *a*_0_, *b*_0_ are the colorimetric parameters of the reference sample according to the CIELab color space (*L*, *a*, and *b* are the parameters of the samples after immersion in the buffered solutions).

Contact angle and drop absorption time evaluations were conducted with EasyDrop (Krüss Scientific GmbH, Hamburg, Germany) using deionized water in order to evaluate the hydrophilic/hydrophobic behavior of the fabrics and further discuss the improved dyeing performances, antibacterial properties, and overall stability to washing of chitosan coating and dyeing.

Fourier transformed infrared (FT-IR) analysis was carried out using the attenuated total feflection (ATR) technique in the range from 4000 to 650 cm^−1^ with 50 scansions and 4 cm^−1^ of band resolution by means of a Thermo Scientific Nicolet iZ10 spectrometer (Thermo Fisher Scientific, Waltham, MA, USA) equipped with a Smart Endurance™ apparatus. The FT-IR spectra were recorded using OMNIC 9 software.

### 2.6. Antibacterial Experiments

The antibacterial activity was evaluated on chitosan-treated samples, according to ASTM E 2149-2013 “Standard test method for determining the antimicrobial activity of antimicrobial agents under dynamic contact conditions”. This method employed Gram-negative *E. coli* and Gram-positive *S. aureus*. Antibacterial tests were performed by diluting the test culture incubated in a nutrient broth (the bacterial inoculum) in a buffer (pH 7.0) to yield a concentration of 1.5–3.0 × 10^5^ CFU/ mL (working dilution).

For each test, a sample of 0.5 g of fabric was immersed into a flask containing 25 mL of the working dilution. All flasks were shaken for different contact times (12 min, 30 min, and 1 h) at 190 rpm at room temperature.

After a series of dilutions with a buffer until a concentration of 150–300 CFU/mL, 1 mL of the liquid was plated in 15 mL of yeast extract agar. The inoculated plates were incubated at 37 °C for 24 h and surviving cells were counted by the plate count method. The tests were conducted in duplicate.

The antibacterial activity was expressed as a percent reduction in the organisms after contact with the test specimen compared to the number of bacterial cells surviving after contact with the control, according to Equation (2):(2)$$\%\;reduction\left(CFU{mL}^{-1}\right)=\frac{B-A}{B}\times 100$$
where A is CFU/mL after contact (end test) and B is CFU/mL at zero contact time (reference).

## 3. Results and Discussion

First, it was necessary to verify whether untreated fabrics have negligible antibacterial effects. Using the ASTM method with both Gram-positive and Gram-negative bacteria, untreated cotton and polyamide 6,6 resulted in a bacterial reduction of 0%.

### 3.1. Chitosan Antibacterial Activity

The mechanism of the antibacterial activity of chitosan is related to its cationic-charged chemical structure, which is influenced by many factors, including the concentration of chitosan, pH value, temperature, degree of deacetylation, cell growth phase, and types of microorganisms [31].

It is well known that Gram-positive and Gram-negative bacteria have different cell wall compositions. In fact, the outer membrane of Gram-negative bacteria comprises essentially lipopolysaccharides containing phosphate and pyrophosphate groups, which renders the bacterial surface a density of negative charges superior to the Gram-positive ones [7].

Chung et al. [32] found a higher inhibitory effect on the Gram-negative bacteria because more chitosan adsorption was observed on their cell surface than on the tested Gram-positive bacteria. More negatively charged cell surfaces had a greater interaction with chitosan due to its cationic nature.

According to the literature, cotton fabrics coated with chitosan have shown interesting results. After 12 min of contact between treated fibers and inoculum bacterium, *E. coli* (Gram-negative bacterium) is immediately more sensitive to the antibacterial action of the biopolymer. There is a bacterial reduction of 99.8% compared to 96.6% of *S. aureus*.

Another essential aspect to consider is the contact time. It was observed that for both bacteria species, a more significant bacterial reduction increases as the contact time between the diluted bacterial inoculum and the cotton fabric treated with chitosan increases, as reported in Table 2.

The same experiment was carried out on the synthetic fiber, polyamide 6,6 treated with chitosan. The results, reported in Table 2, showed that 12 min of contact time between the treated polyamide 6,6 and the bacterial inoculum were enough to achieve a 100% bacterial reduction in both microorganisms studied. Hence, the antibacterial action of the biopolymer on the synthetic fiber was almost immediate. The test results highlighted an excellent antibacterial activity of the synthetic fabric treated with chitosan.

In the specific case of Gram-negative bacterium (i.e., *E. coli*), considering the antibacterial properties of the two different types of fibers, the results showed that the greater antibacterial action was obtained by the synthetic fiber treated with chitosan compared to the natural fiber, considering 12 and 30 min of contact. In both reduced contact times, the antibacterial effect obtained was 100% and the bacterial reduction was 99.8%, considering the natural fiber treated with chitosan, which was in any case an excellent result.

### 3.2. Cross-Linking Properties of Chitosan in Dyeing

Another very interesting property of chitosan is to bind the molecules of some natural dyes and allow for the dyeing of the textile substrate, especially cellulosic ones to which it links very well.

The molecular structure of chitosan can act as a “bridge” between the textile substrate and the dye molecule due to the protonation of nitrogen groups in an acid environment. Generally, synthetic dyes are preferred for dyeing fabrics, and are synthesized ad hoc to be applied to the various types of fibers. Synthetic dyes, however, are a major source of wastewater pollution. Natural dyes are, on the other hand, less toxic and less allergenic thanks to the existence of a large number of structurally different active compounds [25].

The natural polymer is distributed homogeneously on the fabric, as confirmed by dyeing tests using Carmine Red, an acid dye, and by the colorimetric analysis.

### 3.3. Antibacterial Test after Dyeing

Since chitosan is an excellent cross linker to improve the adhesion of natural dyes to fabrics, the antibacterial effect was also tested on fabrics treated with chitosan and dyed with Carmine Red. In this way, it was possible to understand if only the dye had a minimal influence on the antibacterial property conferred by chitosan.

In the following table (Table 3), the results obtained considering the bacterial reduction after the dyeing are reported.

Fabrics (both natural and synthetic) functionalized with chitosan and subsequently dyed with Carmine Red showed good antibacterial properties, especially considering the polyamide 6,6 against Gram-negative bacteria. The dyeing influenced the chitosan antibacterial activity only at a shorter contact time with polyamide 6,6, observing a bacterial reduction of 70.6% at 12 min and 83.2% at 30 min of *S. aureus*.

In general, the bacterial reduction obtained is nearly 100% for both types of fibers with a contact time of 30 min or more. Therefore, the dye had a limited effect on the antibacterial activity of chitosan.

In view of a further exploitation, it is interesting to highlight that the fabrics treated with chitosan, whether they have undergone a dyeing process or not, maintain their antibacterial activity over time.

### 3.4. Durability to Washing

The various finished textile fabrics must have the characteristic of being resistant to washing. Therefore, the fabrics treated with chitosan and dyed were washed [30] and subsequently their antibacterial properties were evaluated to verify the resistance of the treatment.

Almost the same antibacterial behavior was found for both microbial species tested: the antibacterial action of chitosan was maintained at higher contact time. The results reported in Table 4 note that the antibacterial action of chitosan is stronger against Gram-negative bacteria.

Furthermore, the antibacterial activity was reduced only for short contact times (12 and 30 min), while at 1 h, the percentage bacterial reduction remained higher than 90%, and in most cases higher than 95%.

### 3.5. Analysis of Surface Morphology

SEM images in Figure 4 show the morphological aspect of natural and synthetic fibers. Cotton fibers presented a ribbon-like appearance (Figure 4A,B) with a rough surface, while the synthetic polyamide 6,6 fibers have a circular section with a more smooth surface (Figure 4C,D).

Figure 4A,C showed a homogeneous distribution of chitosan on both cotton and polyamide 6,6 fibers.

An important aspect to consider was that washing does not remove the chitosan; the biopolymer was resistant to repeated laundering both on cotton (B) and polyamide 6,6 (D) fabrics. In fact, good antibacterial results were obtained on dyed and washed chitosan-treated samples.

### 3.6. Colorimetric Analysis

Colorimetric analysis was performed on cotton and polyamide 6,6 fabrics. For each sample, the average CIELab ∆*E* values were calculated on three different measurements and the results are reported in Table 5.

As shown in Table 5, both the natural and synthetic chitosan-treated fabrics had an intense coloration (while the untreated fabrics cannot be dyed, as shown in Figure 2 and Figure 3). The dyed and washed samples had a ∆*E* value slightly lower than the freshly dyed fabrics. This result was due to the washing effect because the washing removed weakly linked dye molecules, and therefore reduced the dyeing intensity. However, the dyed and washed samples were brighter than the freshly dyed fabrics because they were more similar to the reference sample (untreated cotton and polyamide 6,6 fabrics).

On the other hand, dyed chitosan-treated fabrics showed a higher ∆*E* value compared to the “white” because they had not been subjected to washing, maintaining a high degree of dyeing intensity.

Therefore, colorimetric analysis demonstrated the resistance of dyeing natural and synthetic fibers due to the presence of chitosan that was being distributed homogeneously on the fabric, ensuring dyeing and strong cross-linking properties.

### 3.7. Contact Angle

Contact angle measurements were performed using water in order to evaluate the hydrophilic/hydrophobic behavior of the fabrics. Since some samples absorbed the water drop quickly, a robust measure of the contact angle was impossible. In this case, only the absorption time was measured. The results are reported in Table 6 and Table 7.

Untreated cotton fabric showed extremely high hydrophilicity since it could absorb a drop of water in about 100 ms; therefore, the contact angle was impossible to measure. The same situation was found on cotton treated with chitosan. In this case, the presence of the biopolymer onto the fabric increased the absorption time to 1 s. A contact angle of 121.3 ± 4.1° was measured on the dyed chitosan-coated cotton fabrics (Figure 5A) and the water drops were not absorbed, whereas a decrease in the contact angle to about 95° was observed on the fabrics after washing (Figure 5B) and the fabric was able to absorb water again in about 4 s.

The synthetic fiber showed an important hydrophobicity that made it possible to measure the contact angle. In all the polyamide 6,6 fabrics, the drop of water remained on the fabric surface for a long time without being absorbed. The measurements of contact angle decreased from 137.6 ± 4.3° for untreated polyamide 6,6 (Figure 5C) to 106.7 ± 7.9° for dyed and washed chitosan-treated polyamide 6,6 (Figure 5F).

It is worth noting that the chitosan treatment and dyeing have different effects on the hydrophilicity of the fibers depending on their hydrophilic/hydrophobic nature. Hydrophilic fibers such as cotton become progressively more hydrophobic when treated with chitosan and then dyed. This behavior is likely due to the fact that both the chitosan and dye led to the inaccessibility of water to the hydrophilic groups of cotton as well as inside the fibers.

On the other hand, in a hydrophobic fiber such as polyamide 6,6, the chitosan treatment and the following dyeing added hydrophilic groups to the fibers, making them progressively more hydrophilic. However, the chitosan treatment and dyeing involved just the surface of the fibers and, for that reason, the fabrics were not able to absorb the water drops at any stage.

### 3.8. Infrared Spectroscopy

FT-IR analysis was carried out on the fabrics with the ATR technique. The spectra are shown in Figure 6 compared to the spectrum of pure chitosan powder. The spectrum of chitosan is characterized by a broad absorption band in the range of 3000–3600 cm^−1^ attributed to O–H stretching, a band at 2870 cm^−1^ attributed to C–H stretching, and several overlapping peaks at about 1000 cm^−1^ attributed to C–O stretching and C–O–C bridge [33].

The spectral features of chitosan are similar to those of cotton as shown in Figure 6. A small difference can be found in the range of 1500–1700 cm^−1^, where chitosan has two partially overlapping weak adsorption bands attributed to the C=O stretching (Amide I) at 1655 cm^−1^ and N–H bending at 1560 cm^−1^ (Amide II), while cotton has a single peak attributed to Amide I and H–O–H bending of absorbed water at 1650 cm^−1^ [34,35].

No significant differences are noticeable in the spectra related to chitosan-treated cotton fabrics compared to the untreated cotton, except for the peak reduction at 1650 cm^−1^, which is probably due to the desorption of water substituted by the linking of the chitosan. In fact, dyeing and washing did not alter the spectra.

Polyamide 6,6 has a typical peak at 3300 cm^−1^ attributed to N–H stretching. Moreover, the two peaks at 2960 cm^−1^ and 2850 cm^−1^ correspond to CH_2_ bonds, while 1630 cm^−1^ and 1560 cm^−1^ indicate the C=O stretching (Amide I) and N–H bending (Amide II) vibrations [36].

Spectral features of chitosan-treated polyamide 6,6 fabric do not differ from the untreated polyamide 6,6 fabric (Figure 6B). Moreover, the spectra of chitosan-treated polyamide 6,6 fabrics after dyeing and washing do not show significant differences with the untreated fabrics, except for a slight increase in the absorption intensity of around 1000 cm^−1^, where chitosan has the most intense absorption peak (see the box in Figure 6A).

FT-IR analysis revealed that the amount of chitosan linked to both cotton and polyamide 6,6 is little, but enough to exert excellent antibacterial effects and to change the hydrophilic/hydrophobic behavior of both fabrics. This finding is in agreement with SEM observations that have shown a thin coating of the fibers.

## 4. Conclusions

In the present work, cotton and polyamide 6,6 fibers were coated with chitosan, a natural biopolymer to be tested against Gram-positive (*S. aureus*) and Gram-negative (*E. coli*) bacteria using the ASTM standard method that requires 1 h of contact between the bacterial inoculum and the treated fabrics.

Chitosan, a useful non-toxic biopolymer, can be used as an effective antibacterial textile finish and an excellent premordant for dyeing. The innovative part of the work was to quantify the antibacterial efficacy of fabrics functionalized with chitosan at shorter contact times (12 min, 30 min, and 1 h) between the treated fabrics and the bacterial inoculum. In this way, it was possible to evaluate the trend of bacterial reduction over time on this type of material. A chitosan solution was prepared and fabrics were dipped into the solution overnight to promote the adsorption of chitosan on the fabrics.

Both cotton and polyamide 6,6 fibers treated with chitosan proved to be highly antibacterial after only 12 min of contact with the bacterial inoculum. Furthermore, the bactericidal action of the chitosan-treated fabrics is excellent up to 1 h of contact, as expected.

Since chitosan has the property of cross-linking natural dyes to provide fabrics with better dyeing and nonfading, a natural dye known as Carmine Red was used to evaluate the premordant property of chitosan, as well as the antibacterial properties of treated fabrics after dyeing. The results were very satisfactory in both aspects since the dyeing is improved by the treatment with chitosan and the bacterial reductions were nearly 100% after 30 min of contact time on both fibers under study.

The results showed that washing reduced the antibacterial effect, especially for evaluations at short contact times.

It can be concluded that chitosan is a biopolymer with a very strong and fast antibacterial efficacy that does not decrease significantly after dyeing. A decreased antibacterial activity was observed after washing only when short contact times were used in testing.

Finally, from a testing point of view, implementing procedures with different contact times were useful to highlight differences that can allow for discriminating among pairs of material/bacterium, even when excellent biocidal actions were found at a fixed contact time for a specific bacterium.

## Figures and Tables

**Figure 1 jfb-14-00524-f001:**
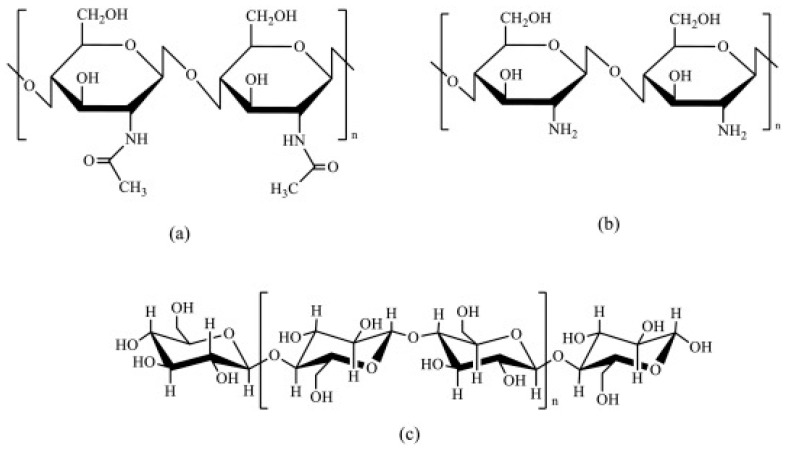
Structure of chitin, (**a**) chitosan, (**b**) and cellulose (**c**).

**Figure 2 jfb-14-00524-f002:**
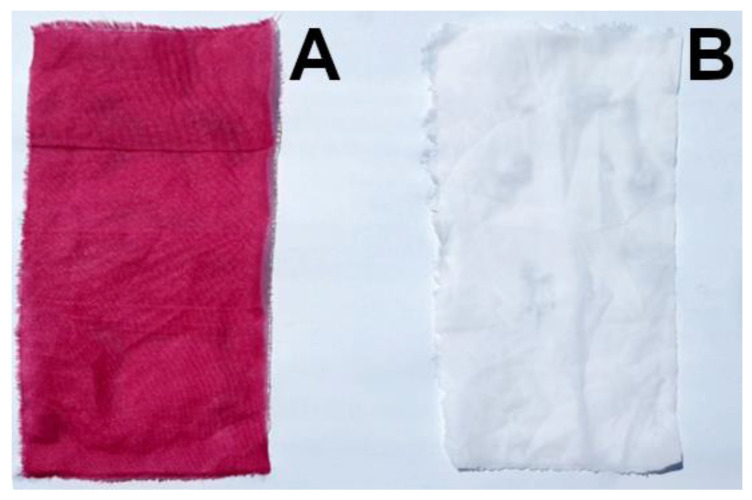
Dyed chitosan-treated cotton fabric (**A**), dyed untreated cotton fabric (**B**).

**Figure 3 jfb-14-00524-f003:**
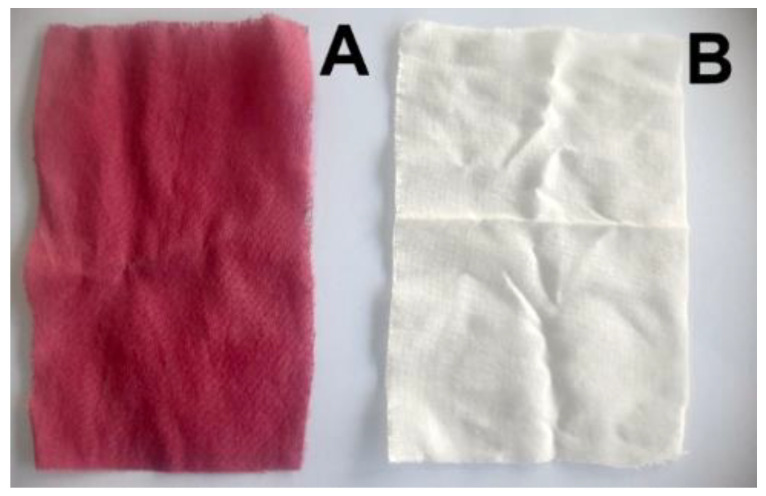
Dyed chitosan-treated polyamide 6,6 fabric (**A**), dyed untreated polyamide 6,6 fabric (**B**).

**Figure 4 jfb-14-00524-f004:**
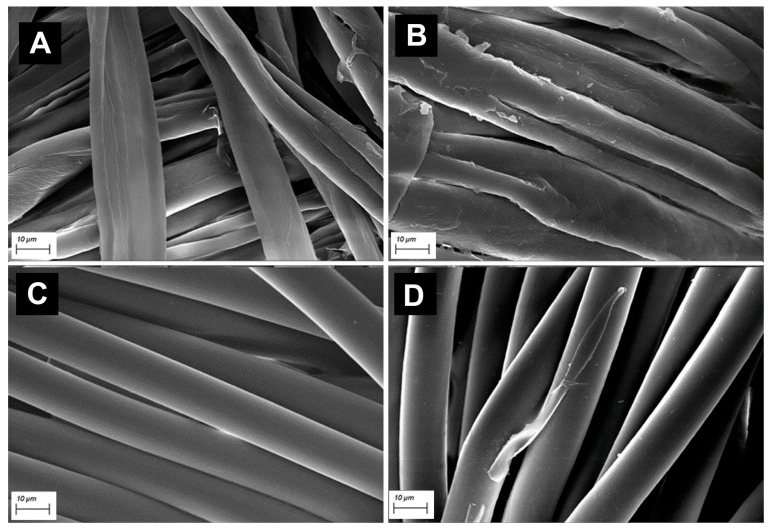
SEM images of cotton fibers treated with chitosan (**A**). Chitosan resistance on Carmine Red dyed and washed cotton fibers (**B**). SEM images of polyamide 6,6 fibers treated with chitosan (**C**). Chitosan resistance on Carmine Red dyed and washed polyamide 6,6 fibers (**D**). Scale bars: 10 µm.

**Figure 5 jfb-14-00524-f005:**
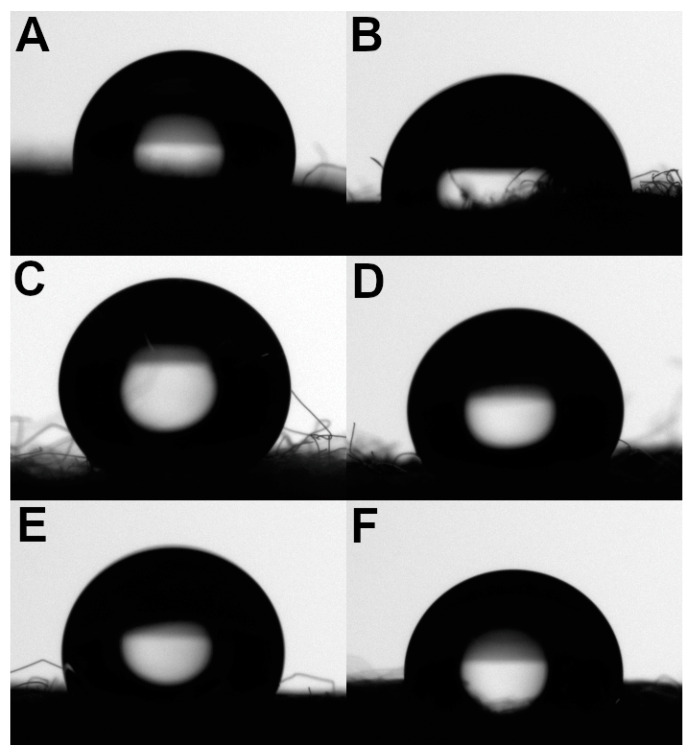
Pictures of the water drops during the contact angle measurements: dyed chitosan-coated cotton (**A**); dyed and washed chitosan-treated cotton (**B**); polyamide 6,6 (**C**); chitosan-coated polyamide 6,6 (**D**); dyed chitosan-coated polyamide 6,6 (**E**); dyed and washed chitosan-treated polyamide 6,6 (**F**).

**Figure 6 jfb-14-00524-f006:**
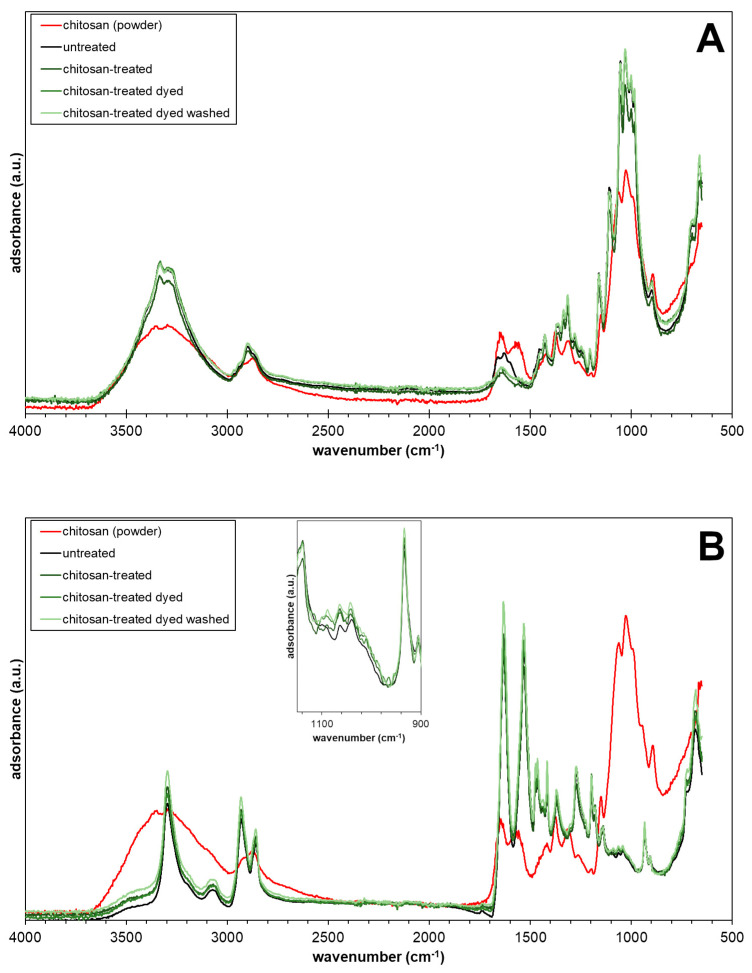
FT-IR spectra of (**A**) cotton and (**B**) polyamide 6,6 fabrics (untreated: black line; fresh chitosan-treated: dark green line; dyed chitosan-treated: green line; dyed chitosan-treated after washing: light green line) compared with chitosan powder (red line).

**Table 1 jfb-14-00524-t001:** Possible toxic effects of some synthetic antimicrobial agents on human beings.

Synthetic Agent	Toxic Effects
QACs	respiratory irritation, nausea, skin, and eye irritation [8]
silver	argyria, contact dermatitis, mucous membrane irritation [9]
zinc pyrithione	developmental and neurotoxicity [10]
azo dyes	carcinogenic [11]
triclosan	endocrine disrupter, skin and eye irritation [12]
zinc oxide	cytotoxicity, apoptosis induction, ROS generation [13]

**Table 2 jfb-14-00524-t002:** Bacterial reduction rate of chitosan-treated cotton and polyamide 6,6 fabrics.

Fabric	Contact Time	Bacterial Reduction (%)	
		*E. coli*	*S. aureus*
	12 min	99.8	96.6
Cotton	30 min	99.8	100
	1 h	100	100
	12 min	100	100
Polyamide 6,6	30 min	100	100
	1 h	100	100

**Table 3 jfb-14-00524-t003:** Bacterial reduction after dyeing of chitosan-treated cotton and polyamide 6,6 fabrics.

Fabric	Contact Time	Bacterial Reduction (%)	
		*E. coli*	*S. aureus*
	12 min	94.2	97.1
Cotton	30 min	97.8	94.2
	1 h	98.5	98.9
	12 min	90.9	70.6
Polyamide 6,6	30 min	99.1	83.2
	1 h	99.7	95.2

**Table 4 jfb-14-00524-t004:** Bacterial reduction after dyeing and washing of chitosan-treated cotton and polyamide 6,6 fabrics.

Fabric	Contact Time	Bacterial Reduction (%)	
		*E. coli*	*S. aureus*
	12 min	59.0	64.4
Cotton	30 min	92.8	84.5
	1 h	98.5	97.7
	12 min	61.4	36.5
Polyamide 6,6	30 min	96.0	47.9
	1 h	95.5	90.4

**Table 5 jfb-14-00524-t005:** ∆*E* values on cotton and polyamide 6,6 fabrics.

Fabric	Chitosan-Treated Cotton		Chitosan-Treated Polyamide 6,6	
	Dyed	Dyed and Washed	Dyed	Dyed and Washed
∆*E*	43.8	41.8	50.5	48.9

**Table 6 jfb-14-00524-t006:** Measurements of contact angle and absorption time on different treated cotton fabrics.

Fabric	Contact Angle (°)	Absorption Time (s)
Cotton	−	0.1
Chitosan-treated cotton	−	1.0
Dyed chitosan-treated cotton	121.3 ± 4.1	∞
Dyed and washed chitosan-treated cotton	95.1 ± 16.4	3.95

**Table 7 jfb-14-00524-t007:** Measurements of contact angle on the different treated polyamide 6,6 fabrics.

Fabric	Contact Angle (°)	Absorption Time (s)
Polyamide 6,6	137.6 ± 4.3	∞
Chitosan-treated polyamide 6,6	122.5 ± 18.2	∞
Dyed chitosan-treated polyamide 6,6	118.8 ± 5.7	∞
Dyed and washed chitosan-treated polyamide 6,6	106.7 ± 7.9	∞

## Data Availability

Data sharing not applicable.

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
