# Peer review of "Dyeing Improvement and Stability of Antibacterial Properties in Chitosan-Modified Cotton and Polyamide 6,6 Fabrics"

_jfb, 2023, doi:10.3390/jfb14100524_

Round 1

Reviewer 1 Report

Dear authors, 

Concerning the manuscript “Dyeing improvement and stability of antibacterial properties in chitosan-modified cotton and polyamide fabrics” by M. Piccioni and others

There is no doubt that it summarizes valuable information concerning using chitosan as bio-cationic agent for treated cotton and polyamide 66 fabrics to improving some functional properties and dyeability for natural dye.

The work seems quite organized, however, I find that there are some aspects that the authors could consider, in order to facilitate the reading and understanding of the article, such as:

1.     It would be advisable to change the title.

2.     In all This manuscript please change term “polyamide” to “polyamide 66”.

3.     The abstract need to be clear and significantly of application and more scientific because missing details and general. please need to reorganize all abstract.

4.     In keywords, please add polyamide 66.

5.     In introduction, the background is long, the authors should enrich this part and emphasize the necessity of "application of different agents " for modification of this fabrics, give some examples and need more focus of bio-cationic agents.

6.     In introduction, recheck and recognize the 3 last paragraphs. also, the objective is unclear.

  1. This manuscript is organized but lacks specific comparative analysis. What are the advantages of "bio-cationic agents" compared disadvantages for treatment these fabrics?

8.     The methodology must be clearer for experimental, because missing fabric composition and chemicals used for treatment.

9.     In experimental, line 134-136 was wrong pleas recheck ISO 105-F02 and ISO 105-F03. this test methods for Textiles — Tests for color fastness (missing in results and discussion section).

10. In experimental, line 142-146 his method not design for fabric treatment please look at pad dry cure methods ref https://doi.org/10.3390/polym14194211 and redesign method.

11. In experimental, Carmine Red missing chemical structure and color index and main components. For more details for discussion Schematic representation of the interactions between cochineal dye and wool fiber look at ref https://doi.org/10.1007/s12221-017-6923-3

12. line 155-163, need to recognize it because it needs more explanation, missing auxiliaries used and the role of it, dyeing curve and conditions.

13. This manuscript needs some tests method must be add such as N% content, physical (mechanical) properties, vapor water sorption, EDX, XPS, durably to wash K/S, fastness properties (dry and wet Rubbing, Perspiration and Hot Pressing), Dye Fixation Measurement, Adsorption Isotherms, FTIR, UV-Vis, and dye bath absorption.

14. Some test method mentions not right like ISO 105-CO6(2010). Author mention that five stainless steel balls used but it is wrong. Please, recheck all test methods and details.

15. Recheck figures and tables titles, details, conditions, font size and clear label.

16. In discussion and perspectives, the author should consider giving some methodological design about how to improve the performance of using this process more specifically.

17. In discussion and perspectives, please need more explanation of role of bio-cationic agents used and role of binding agent (need more explains about reactions and side effect).

18. In all manuscript, change term Pristine to untreated.

19. Line 284-285, need more explanation.

20. Line 296-308,, need more explanation and why acid dye?

21. 3.5.SEM Observations, title must be change.

22. Line 449-459, need more explanation and why use this test method specifically.

23. In conclusions and perspectives, please reorganize and add the optimum conditions of treatment with full details. (the optimum conditions is missing and details, repeating the sentences and please focus in objective).

26. Generally, the dye used had functionalization properties when applied for cotton and nylon 66 or not?

1.Please revisit the entire manuscript for minor grammar issues.

2. Please check the typo error, font size, and spelling

Author Response

Point-by-point response to Comments and Suggestions for Authors

Comments 1: It would be advisable to change the title.

Response 1: Unfortunately, we can’t follow this suggestion because the Reviewer didn’t explain how to change the title. In any case, we think that the title is suited to describe the research work.

Comments 2: In all This manuscript please change term “polyamide” to “polyamide 66”.

Response 2: Thanks for the suggestion. We agreed to change the term “polyamide” to “polyamide 6,6” to be more specific with the terminology. We changed in all the text the word polyamide with polyamide 6,6

Comments 3: The abstract need to be clear and significantly of application and more scientific because missing details and general. please need to reorganize all abstract.

Response 3: We agreed with this comment. After the revision, the abstract contents underlines that chitosan is a natural biopolymer and detailed the use of two different bacteria strains for the antibacterial tests, as well as the use of Carmine Red as a natural dye. We also specified the standard method applied.

Comments 4: In keywords, please add polyamide 66.

Response 4: We added polyamide 6,6 as keyword in the article.

Comments 5: In introduction, the background is long, the authors should enrich this part and emphasize the necessity of "application of different agents " for modification of this fabrics, give some examples and need more focus of bio-cationic agents.

Response 5: We thank the Reviewer for the suggestion. We emphasized the advantages to apply natural biopolymers. We added examples in the use of compounds extracted from different parts of plants (e.g. terpenoids, tannins) that represent a source of natural antimicrobials. Examples of bio-mordant applications of chitosan in textile have been provided in the introduction.

Comments 6: In introduction, recheck and recognize the 3 last paragraphs. also, the objective is unclear.

Response 6: The objective of the paper in the introduction has been rewritten to be more clear. Since the existing test methods suggest to use 1 hour of contact times between bacterial inoculum and antibacterial specimen of 1 hour or more, this work was evaluating if shorter contact times can be more selective in evaluating antibacterial  performances of fabrics of natural and synthetic fibers treated with chitosan, also after dyeing and washing.

Comments 7: This manuscript is organized but lacks specific comparative analysis. What are the advantages of "bio-cationic agents" compared disadvantages for treatment these fabrics?

Response 7: The advantages of “bio-cationic agents” (i.e., chitosan) are represented by their chemical structure which carrying positively charged amine groups that interacts, through an electrostatic interaction mechanism, with the negative charges of teichoic acid (Gram-positive) and lipopolysaccharide (Gram-negative), leading to bacterial death. Moreover, the layer of chitosan enhances dyeing performances and stability.

Comments 8: The methodology must be clearer for experimental, because missing fabric composition and chemicals used for treatment.

Response 8: In the Section 2.1 “Bacteria strains and fabrics” is reported the use of two types of fiber: cotton fabric (suitable for ISO 105-F02, mass per unit 110 g/m2) and polyamide 6,6 fabric (suitable for ISO 105-F03, mass per unit 130 g/m2).

The chemicals were listed in the Sections where the process is described.

Comments 9: In experimental, line 134-136 was wrong pleas recheck ISO 105-F02 and ISO 105-F03. this test methods for Textiles — Tests for color fastness (missing in results and discussion section).

Response 9: The fabrics used follow the requirements as adjacent fabrics for the standards test methods related to the washing fastness, i.e. ISO 105 F02 “Specification for cotton and viscose adjacent fabrics” and ISO 105 F03 “Specification for polyamide adjacent fabric”, as reported in the Section 2.1.

Comments 10: In experimental, line 142-146 his method not design for fabric treatment please look at pad dry cure methods ref https://doi.org/10.3390/polym14194211 and redesign method.

Response 10: The pad-dry-cure process indicated in ref https://doi.org/10.3390/polym14194211 differs from our methods just for the impregnation time and wet pick-up value. The principle is the same.   

Comments 11: In experimental, Carmine Red missing chemical structure and color index and main components. For more details for discussion Schematic representation of the interactions between cochineal dye and wool fiber look at ref https://doi.org/10.1007/s12221-017-6923-3

Response 11: We agree with this comment. We added more information about Carmine Red dye.

Comments 12: line 155-163, need to recognize it because it needs more explanation, missing auxiliaries used and the role of it, dyeing curve and conditions.

Response 12: For the dyeing process we didn’t use auxiliaries. An Ahiba Nuance Top Speed II (Datacolor Italia srl, Italy) dyeing machine was used. The process is described in details in the article. The dyeing temperature setting is reported in the article: “The working temperature was set at 100 °C with a 1 °C/min heating rate and left for 1 h”.

Comments 13: This manuscript needs some tests method must be add such as N% content, physical (mechanical) properties, vapor water sorption, EDX, XPS, durably to wash K/S, fastness properties (dry and wet Rubbing, Perspiration and Hot Pressing), Dye Fixation Measurement, Adsorption Isotherms, FTIR, UV-Vis, and dye bath absorption.

Response 13: The article has the aim to underline the antibacterial properties of chitosan-treated fabrics at different contact times with bacteria. Unfortunately, we didn’t have the listed techniques available. We agree to add FT-IR analysis. On the other hand, the manuscript already reported results of colorimetry, SEM analysis, and contact angle measurements.

Comments 14: Some test method mentions not right like ISO 105-CO6(2010). Author mention that five stainless steel balls used but it is wrong. Please, recheck all test methods and details.

Response 14: Sorry for the mistake, we used ten stainless steel balls as provide ISO 105-C06 (2010).

Comments 15: Recheck figures and tables titles, details, conditions, font size and clear label.

Response 15: Thanks for the suggestion. We checked figures and tables.

Comments 16: In discussion and perspectives, the author should consider giving some methodological design about how to improve the performance of using this process more specifically.

Response 16: We think that the process (“antibacterial experiments”) is easily applicable and repeatable because the only variable presented is the contact time, the methodology is repeatable in all the passages following the standard method test.

Comments 17: In discussion and perspectives, please need more explanation of role of bio-cationic agents used and role of binding agent (need more explains about reactions and side effect).

Response 17: In the discussion of the Section 3.1 “Chitosan antibacterial activity” we underlined that the chitosan’s mechanism action is influenced by many factors, especially the different cell wall compositions of bacteria. Antibacterial activity of chitosan is more effective against Gram-negative bacteria because they present a major density of negative charges onto the cell wall because lipopolysaccharides. This results in a larger interaction with chitosan’s positive charges.

Comments 18: In all manuscript, change term Pristine to untreated.

Response 18: In page 11, we changed the term “pristine” to “untreated” as suggested.

Comments 19: Line 284-285, need more explanation.

Response 19: The sentence was unclear. We are sorry about that. We rewrite it as follows: “The results, reported in Table 2, showed that 12 minutes of contact time be-tween treated polyamide 6,6 and the bacterial inoculum were enough to achieve a 100 % bacterial reduction on both microorganisms studied.”

Comments 20: Line 296-308,, need more explanation and why acid dye?

Response 20: Carmine Red is an acid dye derived from carminic acid.

Comments 21: 3.5.SEM Observations, title must be change.

Response 21: I changed the title of 3.5 paragraph with “Analysis of surface morphology”

Comments 22: Line 449-459, need more explanation and why use this test method specifically.

Response 22: Water contact angle and drop adsorption time measurements were used to evaluate the hydrophilic/hydrophobic behavior of the fabrics to further discuss improved dyeing performances, antibacterial properties and stability to washing. This sentence has been added in the description of the method in Section 2.5 Characterizations.

Comments 23: In conclusions and perspectives, please reorganize and add the optimum conditions of treatment with full details. (the optimum conditions is missing and details, repeating the sentences and please focus in objective).

Response 23: Thank you for the suggestion. The Conclusions section has been improved. We would like to point out that this study does not concern finding optimum conditions of the treatment, but a comparison between antibacterial testing conditions to be more selective for a fine evaluation of efficient antibacterial fabrics.

Comments 24: Generally, the dye used had functionalization properties when applied for cotton and nylon 66 or not?

Response 24: Cotton and polyamide 6,6 fibers treated with chitosan had functionalization properties because chitosan had cross-linking properties, so it favors the attachment of the dye on the fiber. The dye alone dod not have affinity properties in fact if the untreated fabrics were submitted to dyeing process, at the end the dye was lost (see Figure 2 and Figure 3).

Reviewer 2 Report

The article exhibits strong writing throughout. Nevertheless, there is room for further conciseness in the introduction. Additionally, the conclusion appears to reiterate the results. To enhance the overall quality, it is advisable for the author to revise both the introduction and conclusion sections before final acceptance.

The article is well-crafted, but it would benefit from improved clarity, especially in the abstract and introduction sections. Ensuring that these sections are more straightforward and reader-friendly would enhance the overall comprehension of the paper. Therefore, we recommend that the author rephrases and simplifies the language in these sections.

Author Response

Point-by-point response to Comments and Suggestions for Authors

Comments and Suggestions for Authors

The article exhibits strong writing throughout. Nevertheless, there is room for further conciseness in the introduction. Additionally, the conclusion appears to reiterate the results. To enhance the overall quality, it is advisable for the author to revise both the introduction and conclusion sections before final acceptance.

Comments on the Quality of English Language

The article is well-crafted, but it would benefit from improved clarity, especially in the abstract and introduction sections. Ensuring that these sections are more straightforward and reader-friendly would enhance the overall comprehension of the paper. Therefore, we recommend that the author rephrases and simplifies the language in these sections.

Response: We agree to simplify the language in the section abstract and introduction to make the manuscript clearer.                                                                                        

Reviewer 3 Report

Chitosan modified fabrics for antibacterial activity has been explored in the literature, the novelty of this work is low. The following issues should be addressed.

(1)   The layout of the introduction should be modified.

(2)   The permeability of the modified fabrics should be tested. The antibacterial performance should be evaluated in detail, for example, the images and CFU of bacterial.

(3)   The mechanism of antibacterial activity of chitosan modified fabric should be discussed in detail.

Author Response

Point-by-point response to Comments and Suggestions for Authors

Comments and Suggestions for Authors

Chitosan modified fabrics for antibacterial activity has been explored in the literature, the novelty of this work is low. The following issues should be addressed.

Comments 1: The layout of the introduction should be modified.

Response 1: We modified the layout of the introduction as suggested.

Comments 2: The permeability of the modified fabrics should be tested. The antibacterial performance should be evaluated in detail, for example, the images and CFU of bacterial.

Response 2: The antimicrobial activity was evaluated at different contact times (namely, 12 and 30 minutes and 1 hour). The antibacterial performances were quantified and expressed as the percentage reduction of the microorganisms after contact with the test specimen compared to the number of bacterial cells surviving after contact with the control, according to Equation (2) in the paper, for each contact time studied The surviving colonies were counted on Petri dishes after proper incubation. Unfortunately, we did not take pictures of the Petri at the counting stage.

Comments 3: The mechanism of antibacterial activity of chitosan modified fabric should be discussed in detail.

Response 3: In the Section 1 “Introduction” the antibacterial action of chitosan on living microorganisms is described in detail. Moreover, the chitosan activity is further discussed in the Section 3.1 “Chitosan antibacterial activity”. In brief, the advantages of “bio-cationic agents” are represented by their chemical structure which carrying positively charged amine groups that interact, through an electrostatic interaction mechanism, with the negative charges of teichoic acid (Gram-positive) and lipopolysaccharide (Gram-negative), leading to bacterial death.

Reviewer 4 Report

The manuscript, "Dyeing improvement and stability of antibacterial properties in chitosan-modified cotton and polyamide fabrics," where authors used carmine red dye on the chitosan-modified samples. To my understanding, the chitosan works a mordant to an extent for the reported dye experiments. The author used these samples to evaluate the antibacterial activity and measured the hydrophobicity, microscopy, and color properties.

Points that authors need to revise. 

1. One of the major missing points in the study is underestimating (underwritten) the influence of the cationic nature of chitosan, as it is positively charged and, therefore, easily interacts with the negatively charged bacterial membrane (thus producing antibacterial activity); therefore, chitosan has an intrinsic antibacterial activity. However, because of its cationic-charged surface, it also produces a synergetic effect with other antibacterial compounds (such as synthetic drugs or naturally derived compounds)  

For more information:

(a) doi.org/10.3390/ma16186076

(b) doi.org/10.3390/mi13081265

2. The paper writing is not organized correctly; aberrations are not fully described or described later in the paper before introducing them—for example, QAC (Quaternary ammonium, compounds). Please check thoroughly to maintain the uniformity of the paper.

3. The author stated, "According to their origin, antimicrobial agents can be divided into natural or synthetic." This line needs to be revised. For example, any biological activity from agents broadly falls into two categories, where the agents can be of synthetic or natural origin. Please add more context to the introduction.

4. In Table 1, where the authors mentioned the toxic effects of antimicrobial agents, only preliminary information is reported. In the current context, where the paper evaluates the antibacterial efficacy of prepared samples (chitosan + dying), Table 1 seems irrelevant. Either remove the table or add more specific or technical information to the table. 

5. The author used the terminology antimicrobial in the manuscript (or in some places), while the reported studies are limited to antibacterial; please be specific.

6. Include a figure showing the dying parameters (temperature, addition sequence of agents, and time duration).

7. Were the figures 2 and 3 taken after the sample washing? Please add figures of fabric (before and after washing).

8.  Was there chitosan antibacterial tested beforehand the samples were tested? a bit confusing; please write it clearly. 

Author Response

Point-by-point response to Comments and Suggestions for Authors

Comments and Suggestions for Authors

The manuscript, "Dyeing improvement and stability of antibacterial properties in chitosan-modified cotton and polyamide fabrics," where authors used carmine red dye on the chitosan-modified samples. To my understanding, the chitosan works a mordant to an extent for the reported dye experiments. The author used these samples to evaluate the antibacterial activity and measured the hydrophobicity, microscopy, and color properties.

Points that authors need to revise.

Comments 1: One of the major missing points in the study is underestimating (underwritten) the influence of the cationic nature of chitosan, as it is positively charged and, therefore, easily interacts with the negatively charged bacterial membrane (thus producing antibacterial activity); therefore, chitosan has an intrinsic antibacterial activity. However, because of its cationic-charged surface, it also produces a synergetic effect with other antibacterial compounds (such as synthetic drugs or naturally derived compounds) 

For more information:

(a) doi.org/10.3390/ma16186076

(b) doi.org/10.3390/mi13081265

Response 1: Thank you for the comment. The mechanism of chitosan is discussed in the paper in the Introduction and in the Section 3.1 “Chitosan antibacterial activity”. In particular, the Introduction reported: “All antibacterial mechanisms described are due to chitosan molecules carrying positively charged amine groups that have an electrostatic interaction with negatively charged cell membranes of microorganisms.”

We further stressed the ionic mechanism of the biocidal action of the chitosan, improving Section 3.1.

Comments 2: The paper writing is not organized correctly; aberrations are not fully described or described later in the paper before introducing them—for example, QAC (Quaternary ammonium, compounds). Please check thoroughly to maintain the uniformity of the paper.

Response 2: Thank you for the suggestion. Abbreviations were checked.

Comments 3: The author stated, "According to their origin, antimicrobial agents can be divided into natural or synthetic." This line needs to be revised. For example, any biological activity from agents broadly falls into two categories, where the agents can be of synthetic or natural origin. Please add more context to the introduction.

Response 3: Thank you for the comment. The statement has been changed as follow: “Often antimicrobial agents for textiles are synthetic.”

Comments 4: In Table 1, where the authors mentioned the toxic effects of antimicrobial agents, only preliminary information is reported. In the current context, where the paper evaluates the antibacterial efficacy of prepared samples (chitosan + dying), Table 1 seems irrelevant. Either remove the table or add more specific or technical information to the table.

Response 4: The table has the aim to underlined the toxic effects of synthetic antimicrobial agents and consequentially focus on the use and application of natural antimicrobial agents, which presents many advantages such as biodegradability and non-toxicity.

Comments 5: The author used the terminology antimicrobial in the manuscript (or in some places), while the reported studies are limited to antibacterial; please be specific.

Response 5: We fully agree with the Reviewer. We substituted “antimicrobial” with “antibacterial” in the text when relevant.

Comments 6: Include a figure showing the dying parameters (temperature, addition sequence of agents, and time duration).

Response 6: Dyeing parameters are already described as text in Section 2.3 “Dyeing”. In our opinion, a figure repeating the same parameters should be superfluous. No additive was used during the dyeing. This information was added to the description.

Comments 7: Were the figures 2 and 3 taken after the sample washing? Please add figures of fabric (before and after washing).

Response 7: The figures 2 and 3 were taken before the sample washing. It could be noted that after the dyeing process, the chitosan-treated cotton and chitosan-treated polyamide 6,6 fibers were homogeneously dyed; on the contrary, the untreated fabrics maintained their natural white color because of the poor affinity of Carmine Red to these fibers. Washing results are expressed as ∆E values reported Table 5. The differences of ∆E values (2 or less) could not be noticeable in a picture.

Comments 8: Was there chitosan antibacterial tested beforehand the samples were tested? a bit confusing; please write it clearly.

Response 8: Chitosan-treated fabrics were tested against Gram-positive and Gram-negative bacterial strains.

Round 2

Reviewer 1 Report

Authors have carefully responded the previous comments to despite improved quality of the manuscript after revision, I still have eight minor comments that should be properly addressed:

1.     Line 144. Add fabric composition (knitted, weaving or non-woven) (raw, sourcing or bleaching or what) and chemicals used for treatment (acetic acid or glacial acetic acid).

2.     Line 144.  Replace cotton fabric (suitable for ISO 105-F02, mass per unit 110 g/m2) to cotton fabric (suitable for ISO 105-F02, mass per unit (115 ± 5) g/m2) and recheck ISO 105-F03.

3.     Line 155.  Must mention pad dry cure method as conventional method for finishing.

4.     Section 2.3. dyeing. please add dyeing curve and change to 2.3. dyeing treated fabrics.

5.      Add full details condition to tables.

6.     3.4 Antibacterial Test after washing Fastness section. Please remove capital from test and fastness (please in all manuscript) and change to 3.4. durability to wash.

7.     3.6 Colorimetric Analysis section. Add K/S or remove this section.

8.     In conclusion section. Please need to summarize it.

Minor editing of English language required

Author Response

For research article

Response to Reviewer 1 Comments

Authors have carefully responded the previous comments to despite improved quality of the manuscript after revision, I still have eight minor comments that should be properly addressed.

Point-by-point response to Comments and Suggestions for Authors

Comments 1: Line 144. Add fabric composition (knitted, weaving or non-woven) (raw, sourcing or bleaching or what) and chemicals used for treatment (acetic acid or glacial acetic acid).

Response 1: Fabrics description from line 144 was improved as follows: “Adjacent cotton fabric (plain weave fabric suitable for ISO 105-F02, mass per unit area 110.75 g/m2 determined in accordance with ISO 3801, supplied by Testfabrics Inc., USA) and adjacent polyamide 6,6 fabric (plain weave fabric suitable for ISO 105-F03, mass per unit area 130.0 g/m2 determined in accordance with ISO 3801, supplied by Testfabrics Inc., USA) were chosen to be used as textile substrates coated with chitosan for evaluating the antibacterial activity of the biopolymer. Fabrics were used as received from the supplier.” . In section 2.2 we added “glacial” acetic acid.

Comments 2: Line 144. Replace cotton fabric (suitable for ISO 105-F02, mass per unit 110 g/m2) to cotton fabric (suitable for ISO 105-F02, mass per unit (115 ± 5) g/m2) and recheck ISO 105-F03.

Response 2: We checked the Certificate of Conformity from the fabrics supplier adding information in the text, see response 1.

Comments 3: Line 155. Must mention pad dry cure method as conventional method for finishing.

Response 3: The pad-dry-cure process indicated in ref https://doi.org/10.3390/polym14194211 differs from our methods for the impregnation time and wet pick-up value. The principle is the same. So it would be wrong to mention the pad-dry-cure process indicated in ref https://doi.org/10.3390/polym14194211

Comments 4: Section 2.3. dyeing. please add dyeing curve and change to 2.3. dyeing treated fabrics.

Response 4: We changed to “2.3 Dyeing” with “2.3 Dyeing of treated fabrics”.

Comments 5: Add full details condition to tables.

Response 5: We checked the manuscript and found that there are all the details condition referred to tables.

Comments 6: 3.4 Antibacterial Test after washing Fastness section. Please remove capital from test and fastness (please in all manuscript) and change to 3.4. durability to wash.

Response 6: We changed in the section 3.4 the title in “durability to washing”.

Comments 7: 3.6 Colorimetric Analysis section. Add K/S or remove this section.

Response 7: We think that ∆E values in section 3.6 are sufficient to describe the colorimetric analysis section for this type of work focused on studying antibacterial activity.

Comments 8: In conclusion section. Please need to summarize it.

Response 8: Conclusion section has been summarized.

Reviewer 3 Report

The revision is satisfying. 

Author Response

Thank you